# A Higher Charlson Comorbidity Index Is a Risk Factor for Hip Fracture in Older Adults During Low-Temperature Periods: A Cross-Sectional Study

**DOI:** 10.3390/medicina60121962

**Published:** 2024-11-28

**Authors:** Ming-Hsiu Chiang, Yi-Jie Kuo, Shu-Wei Huang, Duy Nguyen Anh Tran, Tai-Yuan Chuang, Yu-Pin Chen, Chung-Ying Lin

**Affiliations:** 1Department of General Medicine, Kaohsiung Chang Gung Memorial Hospital, Kaohsiung 833, Taiwan; b101103050@tmu.edu.tw; 2Department of Orthopedics, Wan Fang Hospital, Taipei Medical University, Taipei 116, Taiwan; benkuo5@tmu.edu.tw (Y.-J.K.); ctywfh@tmu.edu.tw (T.-Y.C.); 3Department of Orthopedics, School of Medicine, College of Medicine, Taipei Medical University, Taipei 110, Taiwan; 4Department of Applied Science, National Taitung University, Taitung 950, Taiwan; 111022@w.tmu.edu.tw; 5The International Ph.D. Program in Medicine, College of Medicine, Taipei Medical University, Taipei 110, Taiwan; d142111006@tmu.edu.tw; 6Department of Orthopedics, Faculty of Medicine, Can Tho University of Medicine and Pharmacy, Can Tho 900000, Vietnam; 7Institute of Allied Health Sciences, College of Medicine, National Cheng Kung University, Tainan 701, Taiwan; cylin36933@gs.ncku.edu.tw; 8Department of Occupational Therapy, College of Medicine, National Cheng Kung University, Tainan 701, Taiwan; 9Department of Public Health, National Cheng Kung University Hospital, College of Medicine, National Cheng Kung University, Tainan 701, Taiwan

**Keywords:** older adults, aged, temperature, hip fracture, Charlson Comorbidity Index

## Abstract

*Background and Objectives*: The incidence of hip fractures is increasing, and there have been reports linking cold weather to a higher risk of fractures. This study aimed to evaluate clinical variables in hip fracture patients who may predispose them to such fractures under different temperatures. *Materials and Methods*: This is a cross-sectional study conducted at a single medical center, enrolling older adults (≥60 years) who had experienced a hip fracture. Comprehensive clinical histories and detailed information regarding each patient’s hip fracture were obtained. All meteorological data were extracted from the Taiwan Central Weather Bureau database. Multiple clinical parameters that may have a close connection with the temperature at which the hip fracture occurred were screened. Statistical analysis involved using the Pearson correlation test or the independent Student’s *t* test, followed by generalized estimating equation analysis. *Results*: The cohort comprised 506 older adults with hip fractures. Initial univariate analysis revealed that a history of past cerebrovascular diseases, Charlson Comorbidity Index, patient age, and preinjury Barthel Index were significantly related to the temperature at which the hip fractures occurred. The generalized estimating equation analysis indicated that only the Charlson Comorbidity Index had a considerably inverse association with temperature. This finding suggests that for older adults with a higher Charlson Comorbidity Index, hip fractures tend to occur at lower temperatures and vice versa. *Conclusions*: Comorbidities are the only clinical concern that predisposes older adults to hip fractures under colder temperatures. This epidemiological finding could guide future patient education and hip fracture prevention programs.

## 1. Introduction

The number of hip fractures has been increasing [1]. In Asia, the number of hip fractures is projected to increase from 1,124,060 in 2018 to 2,563,488 in 2050, and the direct cost of treating hip fractures will increase from USD 9.5 to USD 15 billion [2]. In 2018, Taiwan officially became an aged society; the estimated total number of hip fractures in Taiwan is projected to increase from 18,338 in 2010 to 50,421 by 2035 [3]. A better understanding of the epidemiology of hip fractures and their associated risk factors could improve our current preventive measures and help allocate limited healthcare resources more appropriately [4].

Temperature plays a pivotal role in hip fracture risk. Although it may differ by sex and age [5], most studies have found an increase in hip fracture risk with colder temperatures [6,7,8,9]. In countries with extreme temperatures, such as Norway, the increase in hip fracture risk may be exaggerated; Dahl et al. found a 21% higher risk of hip fractures at low temperatures (<0 °C) compared to ≥0 °C [10]. Similar trends were observed irrespective of climate zones; Nishimurathe et al. used a nationwide Japanese inpatient database and concluded that the estimated relative risks of hip fracture were significantly higher for low temperatures and lower for high temperatures [11]. In a retrospective cohort study conducted in a Southern European region, the incidence of hip fractures among a total of 4000 participants was found to be 15% higher during autumn and winter compared to spring and summer [12].

Few studies have investigated patient characteristics that may predispose older adults to hip fractures under low temperatures, apart from reporting hip fracture incidence in different temperature intervals. However, it is critical to identify potential risk factors for hip fractures under extreme temperatures and take preventive measures for these vulnerable populations in advance. Solbakken et al. analyzed a Norwegian national hip fracture database, and the authors found the greatest seasonal variation in hip fracture incidence among the group with low comorbidities, while patients with the most comorbidities showed less seasonal variation [13]. Identifying these risk factors is crucial for implementing targeted preventive measures in vulnerable populations. Therefore, this study aimed to comprehensively evaluate the clinical characteristics associated with hip fractures at different temperature levels in older adults, with a specific focus on the impact of comorbidities, particularly the Charlson Comorbidity Index, on fracture risk in a subtropical monsoon climate region.

## 2. Materials and Methods

### 2.1. Study Population

This is a cross-sectional study conducted in a single medical center in Taipei City, Taiwan. Qualifying patients were men and women aged 60 years or over who had hip fractures, including intracapsular femoral neck fractures and extracapsular (i.e., basal neck, intertrochanteric, or subtrochanteric) fractures. Patients were excluded if they received hip surgery as a result of a condition other than primary hip fracture, including osteoarthritis, trauma, tumor metastasis, infection, and avascular necrosis of the femoral head. Between 23 November 2017 and 23 November 2020, the clinical data of older adults who underwent surgery for hip fracture were collected. Hip fractures were diagnosed based on clinical presentation (e.g., severe hip pain, inability to bear weight) and confirmed through imaging studies, including standard anteroposterior and lateral radiographs of the hip. In cases where the initial X-rays were inconclusive, additional magnetic resonance imaging or computed tomography scans were utilized for definitive diagnosis. Informed consent was obtained from all participants or their caregivers. When consent was received from the patients or their caregivers, interviews were conducted on admission, and basic demographic data, including age, sex, body mass index (BMI), and underlying comorbidities, were collected for analysis by the nursing staff working at the Center of Osteoporosis and Sarcopenia. We specifically recorded the exact timing and location of hip fracture occurrences from the patients themselves or their caregivers. This study was conducted in compliance with the Declaration of Helsinki and was approved by the Ethics Committee of our university on 1 April 2022 (TMU-JIRB N201709053). All participants provided written consent to participate in this study, receive treatments, and publish their data. Deidentification of all patient details was performed soon after the completion of this study. The reporting of this study conforms to the Strengthening the Reporting of Observational Studies in Epidemiology statement (STROBE) [14], outlined in Appendix A.

### 2.2. Bone Mineral Density Assessment and T-Score Determination

The bone mineral density (BMD) of the bilateral hips and lumbar spine was evaluated using dual-energy X-ray absorptiometry (DXA) (GE Healthcare, Madison, WI, USA). Every patient received a DXA examination of these three sites according to the manufacturer’s standard procedures. Scanning was performed by certified radiographers, and the imaging machine was calibrated daily, as per the manufacturer’s recommendations. The T-scores of the hips and lumbar spine were generated by comparing the bilateral hip and lumbar spine BMD values of participants with those of healthy young adults of European descent (the National Health and Nutrition Examination Survey III database).

### 2.3. Measurement of Other Clinical Factors

At the patient’s first emergency department visit for hip fracture, serum blood tests, including complete blood counts, differential counts, albumin, parathyroid hormone, vitamin D, and creatinine levels, were performed. Participants were administered the Barthel Index (BI) [15] by the three authors to evaluate their preinjury performance of activities of daily living. The Chinese version of the BI has been validated with moderate to excellent agreement among raters for individual items (kappa: 0.53–0.94) and for the total score (intraclass correlation coefficient Z = 0.94) [16]. The Charlson Comorbidity Index (CCI) is a validated and easily applicable method used to estimate the risk of death associated with comorbid diseases. It is widely utilized as a predictor of long-term prognosis and survival [17].

### 2.4. Meteorology Data

Taipei city is located in the northern part of Taiwan and geographically falls into the subtropical monsoon climate. Summers are generally hot and humid, lasting from April to the beginning of October. The winters are short and mild, lasting for approximately 3 to 4 months annually. For instance, in 2019, the average temperatures in July and November were 30.3 °C and 19.1 °C, respectively. Snow rarely falls in winter except in some parts of mountain areas. The size of Taipei city is approximately 271.8 km^2^, with limited weather variation across the whole area.

All meteorological data were extracted from the Taiwan Central Weather Bureau database, which is public and can be accessed through the website listed as follows: https://e-service.cwb.gov.tw/HistoryDataQuery/index.jsp (accessed on 10 December 2023). We extracted individual temperature data based on the place and timing of the hip fracture.

### 2.5. Statistics

All statistical analyses were conducted using SPSS Statistics for Windows, version 27.0 (SPSS Inc., Chicago, IL, USA). Categorical variables are presented as frequencies and percentages, and continuous variables are presented as the means ± standard deviations. Missing data or loss of follow-up were excluded from the analysis. Univariate analyses were conducted to identify numerous potential clinical parameters that are significantly associated with the temperature at which the hip fracture occurred. The Pearson correlation test was adopted to evaluate the association between temperature and continuous variables, while the independent Student’s *t*-test was used for categorical variables. Significant factors identified in the univariate analysis were included in a generalized estimating equation (GEE) to screen for the most prominent influencing clinical variables that are related to temperature variation. The GEE model was chosen because it accounts for the correlation between repeated measurements within individuals, which is appropriate given the observational nature of the data and the potential clustering of certain patient characteristics. The GEE approach also accommodates non-normally distributed outcome variables and is robust to missing data when using an exchangeable correlation structure. This model was selected to provide reliable estimates of the association between clinical factors (e.g., Charlson Comorbidity Index) and temperature while adjusting for confounding variables. For all tests, a two-sided *p*-value less than 0.05 was considered statistically significant.

### 2.6. Assistance with Language Editing

AI assistance was utilized to refine and improve the English language quality of this manuscript. Specifically, ChatGPT (developed by OpenAI) was employed to edit sentence structure, enhance readability, and ensure adherence to academic writing standards. All edits were carefully reviewed and verified by the authors to maintain the scientific accuracy and integrity of the manuscript.

## 3. Results

The baseline characteristics of the overall population are presented in Table 1. From 23 November 2017 to 23 November 2020, over 95% of all older adults with hip fractures admitted to the medical center consented to participate in this study. A total of 506 patients with hip fractures, comprising 146 men and 360 women, were recruited. The mean age was 80.8 ± 9.6 years (range from 60 to 103 years). The most prevalent comorbidity was hypertension (65%), followed by type 2 diabetes mellitus (23.5%), malignant cancer history (13%), cerebrovascular diseases (11%), and end-stage chronic kidney diseases (8.7%).

Figure 1 presents the distribution of temperature at which hip fractures occurred and the number of patients. There were 42 hip fracture events that occurred at <15 °C, 136 at 15–19.9 °C, 155 at 20–24.9 °C, 116 at 25–29.9 °C, and 58 at >30 °C. We also presented the number of days with the average daily temperature during the study period as a reference. There were 73 days with temperatures <15 °C, 279 days with temperatures between 15 and 19.9 °C, 321 days with temperatures between 20 and 24.9 °C, 383 days with temperatures between 25 and 29.9 °C, and 31 days with temperatures >30 °C.

In the univariate analysis, the Pearson correlation test revealed a significant association between temperature and patient age, CCI, and preinjury BI (Pearson coefficient = −0.095, *p*-value = 0.03; Pearson coefficient = −0.18, *p*-value = 0.001; and Pearson coefficient = 0.1, *p*-value = 0.001, respectively). The temperature at which the hip fracture occurred was found to be significantly different between patients with a history of cerebrovascular diseases (CVDs) and those without, as determined by an independent Student’s *t*-test (Table 1). We assessed the association between these four clinical parameters and temperature by using GEE. Table 2 shows that among age, CCI, preinjury BI, and CVD history, only CCI was significantly related to temperature. Patients with higher CCI values tended to have hip fractures at lower temperatures and vice versa (β = – 0.48, *p*-value = 0.003).

## 4. Discussion

This cross-sectional study investigated multiple clinical parameters that may have a close connection with the temperature at which hip fracture occurs, first through univariate analysis, followed by GEE analysis. Among the variables investigated, CCI was the only one identified to have a significant association with temperature. Although having multiple comorbidities is a risk factor for hip fractures [18,19,20], our analysis suggests a complex interaction between age, temperature, and comorbidity burden in influencing hip fracture risk. Older adults with higher CCI scores exhibited a greater tendency for hip fractures at lower temperatures, likely due to reduced mobility, impaired thermoregulation, and heightened vulnerability to cold stress. Conversely, patients with lower CCI scores may face increased fracture risk in warmer temperatures, as they are more likely to engage in outdoor activities. This differential effect highlights the need for tailored preventive strategies based on age and comorbidity level, particularly in response to seasonal temperature changes.

Our findings, consistent with previous studies in the literature, suggest that hip fractures in older adults are more likely to occur at lower temperatures, with the CCI being a significant predictor. The increased risk of hip fractures during colder temperatures could be partially explained by environmental hazards, such as icy surfaces, which lead to a higher incidence of falls outdoors. This is particularly relevant in countries with severe winters, like Norway, where Dahl et al. reported a marked increase in hip fracture risk at temperatures below freezing [10]. However, the association between cold temperatures and fractures may not be purely environmental. Physiologically, colder weather may result in decreased muscle function, increased joint stiffness, and reduced reaction times, all of which heighten the risk of falling. Cold-temperature-induced poor physical activity may be a plausible underlying mechanism. Multiple studies have shown that cold temperatures can deteriorate physical performance [21,22], which could further predispose older adults with higher CCI to falling and hip fractures. Interestingly, similar seasonal patterns have been observed for other types of fractures, including wrist and vertebral fractures [23,24], suggesting a broader relationship between temperature changes and fracture risk apart from the hip. This indicates that both environmental and physiological factors may contribute to increased overall vulnerability to fractures in colder weather.

By contrast, the results of this study suggested that older adults with a lower CCI are more likely to suffer a hip fracture at higher temperatures. This may be explained by a different underlying mechanism from the previous one. The study conducted by Kelsey et al. reported an elevated risk for outdoor falls in healthy older adults during walking or engaging in vigorous activities, while frail individuals tended to suffer from indoor falls [25]. The same conclusion was also drawn by Kelekar et al., who found that compared to outdoor falls, the probability of indoor-fall patients having more chronic conditions was higher [26]. In this study, univariate analysis revealed a significant positive relationship between the preinjury BI and temperature. In addition, the preinjury BI was significantly higher in patients who suffered hip fractures outdoors than in those who had hip fractures indoors (83 compared to 96, *p*-value < 0.001). These findings showed that patients who had better physical activity levels tended to experience hip fractures at warmer temperatures and outdoors. In conclusion, older adults with lower CCI may be more susceptible to falling in warmer conditions, such as when participating in outdoor activities, and it was not due to poor physical ability like those who fell under colder conditions.

In the univariate analysis of this study, it was revealed that CVD history was another temperature-related clinical factor that may have an impact on hip fractures. Stroke history is a documented risk factor for hip fracture. A history of stroke increased the overall risk of osteoporosis, which was 1.82-fold higher than that found in a population-based cohort study [27]. A meta-analysis combining the results of 10 observational studies reported that stroke was associated with an approximately twofold increase in hip fracture risk [28], which was similar to the results of another cohort study [29]. Decreased mobility and asymmetric weight bearing in stroke patients, which lead to an imbalance of muscle strength and mass in both paretic and nonparetic sides, were one possible underlying pathogenesis for falling and hip fracture [30]. Hypothermia may lead to excessive clothing that more easily hinders coordination ability, reduces mobility, and causes episodes of falling in older adults with a CVD history.

Temperature could also impact the postoperative prognosis among hip fracture patients, in addition to being a risk factor for hip fracture. Ogawa et al. investigated over 300,000 Japanese older adults with hip fractures and examined their surgical site infection rate. After adjusting for confounders, they reported that the risk of surgical site infection was significantly higher in spring and summer than in winter [31]. In another retrospective study recruiting over 1000 patients, although not statistically significant, it was found that hip fracture surgeries performed in summer had a 10% higher risk of developing postoperative complications compared to patients who had their surgeries during the winter months [32]. A similar trend was found in other studies involving hip arthroplasty and other orthopedic surgeries [33,34,35]. These results remind us that variations in season and environment could have a profound effect on a patient’s overall condition and the course of a disease.

The findings of this study have significant implications for Taiwan’s healthcare system, particularly in light of the rapidly aging population. Given the projected increase in hip fracture incidence, the demand for surgical treatment, rehabilitation, and long-term care is expected to rise. The increased prevalence of comorbidities among older adults, as highlighted by the significant association between CCI and hip fracture risk, underscores the need for targeted prevention strategies. To effectively address this growing healthcare burden, enhanced osteoporosis management, fall prevention programs, and the integration of geriatric care into standard fracture treatment protocols will be crucial. Given the differential risk patterns observed in this study, preventive strategies should be adapted based on the level of comorbidity. For high-comorbidity individuals, interventions should prioritize reducing indoor fall risks, enhancing home safety, and managing osteoporosis, particularly during colder months. By contrast, for low-comorbidity individuals who are at a greater risk of outdoor falls during warmer temperatures, strategies should focus on promoting safe physical activity and using appropriate footwear. These targeted approaches could help mitigate fracture risks across diverse patient groups and varying seasonal conditions.

One strength of our study lies in the fact that we thoroughly reviewed each patient’s clinical history, enabling us to correlate their clinical information with data from the Taiwan Central Weather Bureau database. As a result, we believe our findings reflect a more authentic relationship between temperature and hip fractures than previous studies.

Nevertheless, this study has several limitations. First, selection bias is a common limitation in most observational studies. However, by diligently surveying each admitted participant, we obtained consent from over 95% of the total qualified patients throughout the study period, and we believe that potential selection bias was substantially reduced. Second, the enrollment of only 506 participants might limit the generalizability of the findings, and they may not fully represent the overall condition of older adults with hip fractures in Taiwan. Third, although we recorded the location of falls (indoor vs. outdoor), the temperature data used in our analysis were based on outdoor ambient temperatures from the Taiwan Central Weather Bureau. We did not account for potential differences in indoor climate conditions (e.g., air conditioning in summer, heating in winter), which may differ from the outdoor temperature and could influence the risk of falls. Future research should consider assessing indoor temperature exposure, particularly given the high proportion of indoor falls observed in this study. Additionally, the study design could be improved by including both indoor and outdoor temperature measurements, which would provide a more accurate assessment of the temperature conditions associated with each fracture event. Fourth, we did not record the residential status of each patient (e.g., living independently vs. in a nursing home), which has been demonstrated to be related to hip fracture epidemiology in previous studies [36]. Including this information in future studies could help control for potential confounding factors related to living environments. Fifth, the use of a retrospective design may introduce recall bias, and a prospective cohort approach in future studies would allow for more precise data collection and reduce potential biases. Sixth, we collected detailed clinical data but did not include additional patient characteristics such as physical activity levels, lifestyle factors (e.g., smoking, alcohol use), and comprehensive medication history. These factors could influence fall risk and fracture outcomes and should be considered in future studies to provide a more holistic understanding of risk factors. Lastly, some data on serum vitamin D concentration, T-score, and the parathyroid hormone were missing, which may have influenced the comprehensiveness and interpretation of the study results. Future research should also consider a longer follow-up period and the integration of objective fall risk assessments (e.g., wearable sensors) to enhance the accuracy of risk prediction models and provide more robust conclusions.

## 5. Conclusions

This observational study revealed that the CCI is the only clinical characteristic predictive of hip fracture in response to temperature changes. This knowledge could contribute to future patient education and hip fracture prevention programs, particularly aiding in the efficient allocation of medical resources based on different patient conditions and temperature intervals. The results of this study may also be applied to areas with similar subtropical monsoon climates, such as Shikoku, Jeju Island, and Southeast China. Future studies with larger sample sizes and longer study periods are warranted.

## Figures and Tables

**Figure 1 medicina-60-01962-f001:**
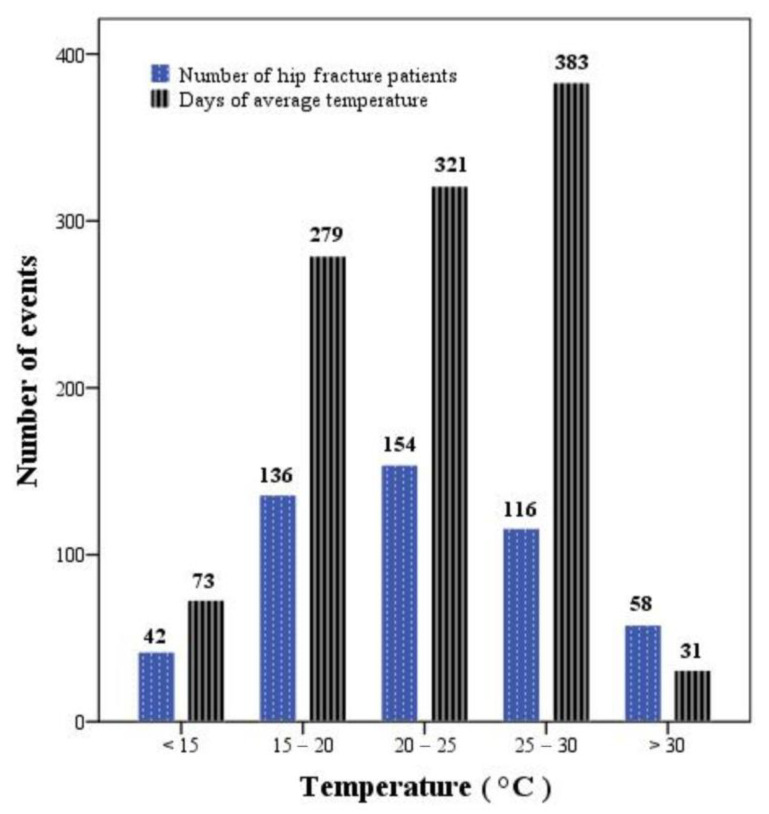
The distribution of hip fracture patients (depicted in blue) and the number of daily average temperatures (depicted in black) are shown in five different temperature intervals during the study period.

**Table 1 medicina-60-01962-t001:** Baseline clinicodemographic characteristics of the study population and their correlation with temperature.

Clinical Characteristics (n = 506)	Mean ± SD/Number (Percentage)	Temperature Correlation *p*-Value (Pearson Coefficient)
Age	80.8 ± 9.6	0.03 (−0.095)
Sex	Male: 146 (29%) Female: 360 (71%)	0.63
Comorbidities		
Hypertension	328 (65%)	0.11
Type 2 diabetes mellitus	119 (24%)	0.08
CKD stage V or ESRD	44 (8.7%)	0.28
Malignant cancer history	65 (13%)	0.21
Cerebrovascular diseases	54 (11%)	0.02
Charlson Comorbidity Index	4.8 ± 1.8	0.001 (−0.18)
BMI	22.3 ± 3.8	0.08
T-score (n = 420)	−3.8 ± 1.1	0.26
Pre-injury Barthel Index	86.3 ± 21.6	0.02 (0.10)
Pre-operation laboratory data		
Hb (gm/dL)	12.1 ± 2.2	0.86
eGFR (mL/min/1.73 m^2^)	76.7 ± 35.9	0.76
Platelet (1000/μL)	207.9 ± 81.3	0.14
WBCs (1000/uL)	12.9 ± 41.3	0.47
PTH (pg/mL) (n = 432)	67.6 ± 203.9	0.39
25(OH)D (n = 364)	20.0 ± 10.4	0.33
Albumin level (g/dL) (n = 461)	3.1 ± 0.9	0.72
Fracture site		
Right	247 (49%)	0.81
Left	259 (51%)	
Fracture type		
Femoral neck fracture	263 (52%)	0.83
Peritrochanteric fracture	243 (48%)	
Falling location (n = 478)	Indoor: 359 (75%)	
	Outdoor: 119 (25%)	

Abbreviations: 25(OH)D: 25-hydroxyvitamin D; BMI: body mass index; eGFR: estimated glomerular filtration rate; WBCs: white blood cells; Hb: hemoglobin; PTH: parathyroid hormone.

**Table 2 medicina-60-01962-t002:** GEE model of factors associated with temperature at the time the hip fracture occurred.

Variables	β	95% CI		*p*-Value
		Lower Limit	Upper Limit	
Age	−0.005	−0.06	0.05	0.87
CCI	−0.48	−0.80	−0.16	0.005
Barthel Index	0.015	−0.01	0.04	0.23
CVDs	−0.91	−2.68	0.83	0.49

Abbreviations: CCI: Charlson Comorbidity Index; CVDs: Cerebrovascular diseases.

## Data Availability

All data generated or analyzed during this study are included in this published article.

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
