# Peer review of "A Higher Charlson Comorbidity Index Is a Risk Factor for Hip Fracture in Older Adults During Low-Temperature Periods: A Cross-Sectional Study"

_medicina, 2024, doi:10.3390/medicina60121962_

Round 1
Reviewer 1 Report
Comments and Suggestions for Authors
This is a well-conducted study and manuscript, so I have only a few minor concerns.
The aim of the study should be stated more clearly at the end of the introduction section.
In the methods, you should state which guidelines they followed. The best one is STROBE (Strengthening the Reporting of Observational Studies in Epidemiology). They should include a STROBE checklist in the supplement.
In Table 1 and throughout the manuscript, I recommend using sex instead of gender because we know that there are only two biological sexes. Gender is more or less a term introduced by western politicians. We are scientists and we have to stick to the evidence and be independent of political influence.
In the results section, I miss a description of the exact distribution of hip fractures. Hip fracture is a very general term. How many femoral neck fractures, pertrochanteric, intertrochanteric, etc?
In the introduction and discussion section, it would be interesting to read why there is a link between temperature changes and hip fractures. Is it because freezing water at 0 degrees leads to more falls? This could explain the findings of the Norwegian study by Dahl et al [10]. Or are there other reasons? Is there the same link with other types of fracture?
Reviewer 2 Report
Comments and Suggestions for Authors
Thank you for the opportunity to evaluate this work. With all sincerity, I consider the theme interesting, original through the studied idea. I congratulate the authors from this point of view. At the same time, I ask the authors to make some clarifications in the text where appropriate in the following directions:
- (lines 82-83) it is indicated that data were collected from the period of 2017-2020 in the month of November. Considering the pandemic period and the chaos of that period (March 2020 and the following months), how were the data collected, were the patients properly treated in a hospital environment or more in an outpatient setting?
- regarding the temperature related to the moment of the hip fracture, did all the patients suffer the respective accidents outside? That is, can the moment of the fracture be correlated with the temperature outside and with the respective season? For the injured patients in the house or indoors, the temperature could have been different, for example in the summer, there would have been air conditioning inside and the temperature would have been low, respectively vice versa in the cold season. Was such an analysis done?
Reviewer 3 Report
Comments and Suggestions for Authors
Thanks for the opportunity to review the paper "Higher Charlson Comorbidity Index is a Risk Factor for Hip Fracture in Older Adults at Lower Temperatures: A Cross-sectional Study"
Abstract
The keywords should not be repeated with the words in the title.
Introduction
Could additional factors, like environmental hazards (ice, snow), also play a role in the relationship between lower temperatures and fracture rates?
Given the projections, how does Taiwan's aging population affect the country’s healthcare system, specifically concerning fracture treatment and prevention? Could you clarify this point?
Materials and Methods
How did the authors address potential selection bias, and could additional patient characteristics have influenced the study outcomes?
Results
In table 2, the decimal places should be standardised.
Could the study expand on how age and temperature interact to affect hip fracture risks for patients with different levels of comorbidities? Could you clarify this point?
Discussion
How could preventive strategies be adjusted to account for both low- and high-comorbidity individuals, considering their different risk patterns? Could you clarify this point?
How could the study design be improved in future research to address these limitations and provide more robust conclusions?
Conclusions
What specific interventions could healthcare providers implement to reduce hip fracture risks based on the findings of this study?
Round 2
Reviewer 3 Report
Comments and Suggestions for Authors
Thanks again for the opportunity to review this paper "Higher Charlson Comorbidity Index is a Risk Factor for Hip Fracture in Older Adults at Lower Temperatures: A Cross-sectional Study"
I was pleased to see the changes made according to the suggestions and comments after the first revision of the document.
The paper is now clearer, more pertinent and more enlightening.
Congratulations on the changes that have improved the paper.